# Representation Learning for Sequence Data with Deep Autoencoding Predictive Components

**Junwen Bai** *
Cornell University
junwen@cs.cornell.edu

**Weiran Wang** †
Google
weiranwang@google.com

**Yingbo Zhou & Caiming Xiong**
Salesforce Research
{yingbo.zhou, cxiong}@salesforce.com

## Abstract

We propose Deep Autoencoding Predictive Components (DAPC) – a self-supervised representation learning method for sequence data, based on the intuition that useful representations of sequence data should exhibit a simple structure in the latent space. We encourage this latent structure by maximizing an estimate of *predictive information* of latent feature sequences, which is the mutual information between the past and future windows at each time step. In contrast to the mutual information lower bound commonly used by contrastive learning, the estimate of predictive information we adopt is exact under a Gaussian assumption. Additionally, it can be computed without negative sampling. To reduce the degeneracy of the latent space extracted by powerful encoders and keep useful information from the inputs, we regularize predictive information learning with a challenging masked reconstruction loss. We demonstrate that our method recovers the latent space of noisy dynamical systems, extracts predictive features for forecasting tasks, and improves automatic speech recognition when used to pretrain the encoder on large amounts of unlabeled data. [1]

## 1 Introduction

Self-supervised representation learning methods aim at learning useful and general representations from large amounts of unlabeled data, which can reduce sample complexity for downstream supervised learning. These methods have been widely applied to various domains such as computer vision (Oord et al., 2018; Hjelm et al., 2018; Chen et al., 2020; Grill et al., 2020), natural language processing (Peters et al., 2018; Devlin et al., 2019; Brown et al., 2020), and speech processing (Schneider et al., 2019; Pascual et al., 2019b; Chung & Glass, 2020; Wang et al., 2020; Baevski et al., 2020). In the case of sequence data, representation learning may force the model to recover the underlying dynamics from the raw data, so that the learnt representations remove irrelevant variability in the inputs, embed rich context information and become predictive of future states. The effectiveness of the representations depends on the self-supervised task which injects inductive bias into learning. The design of self-supervision has become an active research area.

One notable approach for self-supervised learning is based on maximizing mutual information between the learnt representations and inputs. The most commonly used estimate of mutual information is based on contrastive learning. A prominant example of this approach is CPC (Oord et al., 2018), where the representation of each time step is trained to distinguish between positive samples which are inputs from the near future, and negative samples which are inputs from distant future or other sequences. The performance of contrastive learning heavily relies on the nontrivial selection

---

*Work done during an internship at Salesforce Research.

†Work done while Weiran Wang was with Salesforce Research.

[1]Code is available at https://github.com/JunwenBai/DAPC.

of positive and negative samples, which lacks a universal principle across different scenarios (He et al., 2020; Chen et al., 2020; Misra & Maaten, 2020). Recent works suspected that the mutual information lower bound estimate used by contrastive learning might be loose and may not be the sole reason for its success (Ozair et al., 2019; Tschannen et al., 2019).

In this paper, we leverage an estimate of information specific to sequence data, known as *predictive information* (PI, Bialek et al., 2001), which measures the mutual information between the past and future windows in the latent space. The estimate is exact if the past and future windows have a joint Gaussian distribution, and is shown by prior work to be a good proxy for the true predictive information in practice (Clark et al., 2019). We can thus compute the estimate with sample windows of the latent sequence (without sampling negative examples), and obtain a well-defined objective for learning the encoder for latent representations. However, simply using the mutual information as the learning objective may lead to degenerate representations, as PI emphasizes simple structures in the latent space and a powerful encoder could achieve this at the cost of ignoring information between latent representations and input features. To this end, we adopt a masked reconstruction task to enforce the latent representations to be informative of the observations as well. Similar to Wang et al. (2020), we mask input dimensions as well as time segments of the inputs, and use a decoder to reconstruct the masked portion from the learnt representations; we also propose variants of this approach to achieve superior performance.

Our method, Deep Autoencoding Predictive Components (DAPC), is designed to capture the above intuitions. From a variational inference perspective, DAPC also has a natural probabilistic interpretation. We demonstrate DAPC on both synthetic and real datasets of different sizes from various domains. Experimental results show that DAPC can recover meaningful low dimensional dynamics from high dimensional noisy and nonlinear systems, extract predictive features for forecasting tasks, and obtain state-of-the-art accuracies for Automatic Speech Recognition (ASR) with a much lower cost, by pretraining encoders that are later finetuned with a limited amount of labeled data.

## 2 METHOD

The main intuition behind Deep Autoencoding Predictive Components is to maximize the predictive information of latent representation sequence. To ensure the learning process is tractable and non-degenerate, we make a Gaussian assumption and regularize the learning with masked reconstruction. In the following subsections, we elaborate on how we estimate the predictive information and how we design the masked reconstruction task. A probabilistic interpretation of DAPC is also provided to show the connection to deep generative models.

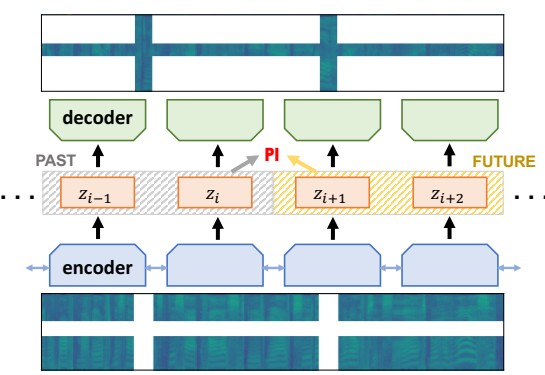

Figure 1: The overall framework of DAPC.

### 2.1 PREDICTIVE INFORMATION

Given a sequence of observations $X = \{x_1, x_2, ...\}$ where $x_i \in \mathbb{R}^n$, we extract the corresponding latent sequence $Z = \{z_1, z_2, ...\}$ where $z_i \in \mathbb{R}^d$ with an encoder function $e(X)$, e.g., recurrent neural nets or transformers (Vaswani et al., 2017).[2] Let $T > 0$ be a fixed window size, and denote $Z_t^{past} = \{z_{t-T+1}, ..., z_t\}$, $Z_t^{future} = \{z_{t+1}, ..., z_{t+T}\}$ for any time step $t$. The predictive information (PI) is defined as the mutual information (MI) between $Z_t^{past}$ and $Z_t^{future}$:

$$MI(Z_t^{past}, Z_t^{future}) = H(Z_t^{past}) + H(Z_t^{future}) - H(Z_t^{past}, Z_t^{future}) \qquad (1)$$

---

[2]In this work, the latent sequence has the same length as the input sequence, but this is not an restriction; one can have a different time resolution for the latent sequence, using sub-sampling strategies such as that of Chan et al. (2016).

where $H$ is the entropy function. Intuitively, PI measures how much knowing $Z_t^{past}$ reduces the uncertainty about $Z_t^{future}$ (and vice versa). PI reaches its minimum value 0 if $Z_t^{past}$ and $Z_t^{future}$ are independent, and it is maximized if $Z_t^{future}$ is a deterministic function of $Z_t^{past}$. Different from the MI estimate used by contrastive learning, which measures the MI between representation at each *single* time step and its future inputs, predictive information measures the MI between two windows of $T$ time steps collectively. The window size $T$ used in PI estimation reflects the time resolution for which the time series is more or less stationary.

PI was designed as a general measure of the complexity of underlying dynamics which persists for a relatively long period of time (Li & Vitányi, 2008). Furthermore, PI is aware of temporal structures: different dynamics could lead PI to converge or diverge even if they look similar. These virtues of PI contribute to the versatility of this measure. The use of PI beyond a static complexity measure (and as a learning objective) is done only recently in machine learning by Clark et al. (2019), which proposes to learn a linear dimensionality reduction method named Dynamical Component Analysis (DCA) to maximize the PI of the projected latent sequence.

One approach for estimating PI is through estimating the joint density $P(Z_t^{past}, Z_t^{future})$, which can be done by density estimation methods such as $k$-NN and binning (Dayan & Abbott, 2001; Kraskov et al., 2004). However, such estimates heavily rely on hyperparameters, and it is more challenging to come up with differentiable objectives based on them that are compatible with deep learning frameworks. Our approach for estimating PI is the same as that of DCA. Assume that every $2T$ consecutive time steps $\{z_{t-T+1}, ..., z_t, ..., z_{t+T}\}$ in the latent space form a stationary, multivariate Gaussian distribution. $\Sigma_{2T}(Z)$ is used to denote the covariance of the distribution, and similarly $\Sigma_T(Z)$ the covariance of $T$ consecutive latent steps. Under the stationarity assumption, $H(Z_t^{past})$ remains the same for any $t$ so we can omit the subscript $t$, and $H(Z^{past})$ is equal to $H(Z^{future})$ as well. Using the fact that $H(Z^{past}) = \frac{1}{2} \ln(2\pi e)^{dT} |\Sigma_T(Z)|$, PI for the time series $z$ reduces to

$$I_T = MI(Z^{past}, Z^{future}) = \ln |\Sigma_T(Z)| - \frac{1}{2} \ln |\Sigma_{2T}(Z)|. \tag{2}$$

Detailed derivations can be found in Appendix A. It is then straightforward to collect samples of the consecutive $2T$-length windows and compute the sample covariance matrix for estimating $\Sigma_{2T}(Z)$. An empirical estimate of $\Sigma_T(Z)$ corresponds to the upper left sub-matrix of $\Sigma_{2T}(Z)$. Recall that, under the Gaussian assumption, the conditional distribution $P(Z^{future}|Z^{past})$ is again Gaussian, whose mean is a linear transformation of $Z^{past}$. Maximizing $I_T$ has the effect of minimizing the entropy of this conditional Gaussian, and thus reducing the uncertainty of future given past.

Though our estimation formula for PI is exact only under the Gaussian assumption, it was observed by Clark et al. (2019) that the Gaussian-based estimate is positively correlated with a computationally intensive estimate based on non-parametric density estimate, and thus a good proxy for the full estimate. We make the same weak assumption, so that optimizing the Gaussian-based estimate improves the true PI. Our empirical results show that representations learnt with the Gaussian PI have strong predictive power of future (see Sec 4.2). Furthermore, we find that a probabilistic version of DAPC (described in Sec 2.3) which models $(Z^{past}, Z^{future})$ with a Gaussian distribution achieves similar performance as this deterministic version (with the Gaussian assumption).

We now describe two additional useful techniques that we develop for PI-based learning.

**Multi-scale PI** One convenient byproduct of this formulation and estimation for $I_T$ is to reuse $\Sigma_{2T}(Z)$ for estimating $I_{T/2}$, $I_{T/4}$ and so on, as long as the window size is greater than 1. Since the upper left sub-matrix of $\Sigma_{2T}(Z)$ approximates $\Sigma_T(Z)$, we can extract $\Sigma_T(Z)$ from $\Sigma_{2T}(Z)$ without any extra computation, and similarly for $\Sigma_{T/2}(Z)$. We will show that multi-scale PI, which linearly combines PI at different time scales, boosts the representation quality in ASR pretraining.

**Orthogonality penalty** Observe that the PI estimate in (2) is invariant to invertible linear transformations in the latent space. To remove this degree of freedom, we add the penalty to encourage latent representations to have identity covariance, so that each of the $d$ latent dimensions will have unit scale and different dimensions are linearly independent and thus individually useful. This penalty is similar to the constraint enforced by deep canonical correlation analysis (Andrew et al., 2013), which was found to be useful in representation learning (Wang et al., 2015).

## 2.2 Masked Reconstruction and Its Shifted Variation

The PI objective alone can potentially lead to a degenerate latent space, when the mapping from input sequence to latent sequence is very powerful, as the latent representations can be organized in a way that increases our PI estimate at the cost of losing useful structure from the input. This is also observed empirically in our experiments (see Sec 4.1). To regularize PI-based learning, one simple idea is to force the learnt latent representations to be informative of the corresponding input observations. For this purpose, we augment PI-based learning with a masked reconstruction task.

Masked reconstruction was first proposed in BERT (Devlin et al., 2019), where the input text is fed to a model with a portion of tokens masked, and the task is to reconstruct the masked portion. Wang et al. (2020) extended the idea to continuous vector sequence data (spectrograms). The authors found that randomly masking input dimensions throughout the sequence yields further performance gain, compared to masking only consecutive time steps. We adopt their formulation in DAPC to handle continuous time series data.

Given an input sequence $X$ of length $L$ and dimensionality $n$, we randomly generate a binary mask $M \in R^{n \times L}$, where $M_{i,j} = 0$ indicates $X_{i,j}$ is masked with value 0 and $M_{i,j} = 1$ indicates $X_{i,j}$ is kept the same. We feed the masked inputs to the encoder $e(\cdot)$ to extract representations (in $R^d$) for each time step, and use a feed-forward network $g(\cdot)$ to reconstruct the masked input observations. $e(\cdot)$ and $g(\cdot)$ are trained jointly. The masked reconstruction objective can be defined as

$$R = ||(1 - M) \odot (X - g(e(X \odot M)))||^2_{fro}. \tag{3}$$

Figure 1 gives an illustration for the masked spectrogram data. We randomly generate $n_T$ time masks each with width up to $w_T$, and similarly $n_F$ frequency masks each with width up to $w_F$. In our experiments, we observe that input dimension masking makes the reconstruction task more challenging and yields higher representation quality. Therefore, this strategy is useful for general time series data beyond audio.

We introduce one more improvement to masked reconstruction. Standard masked reconstruction recovers the masked inputs for the same time step. Inspired by the success of Autoregressive Predictive Coding (Chung & Glass, 2020), we propose a shifted variation of masked reconstruction, in which the latent state $z_i$ is decoded to reconstruct a future frame $x_{i+s}$ (than $x_i$). Formally, the shifted masked reconstruction loss $R_s$ is defined as

$$R_s = ||(1 - M^{\to s}) \odot (X^{\to s} - g(e(X \odot M)))||^2_{fro} \tag{4}$$

where $\to s$ indicates right-shifting $s$ time frames while the input dimensions remain unchanged. When $s = 0$, $R_s$ reduces to the standard masked reconstruction objective, and in the ASR experiments we find that a nonzero $s$ value helps. We ensure no information leakage by enforcing that the portion to be reconstructed is never presented in the inputs. As indicated by Chung et al. (2019b), predicting a future frame encourages more global structure and avoids the simple inference from local smoothness in domains like speech, and therefore helps the representation learning.

To sum up, our overall loss function is defined as the combination of the losses described above:

$$\min_{e,g} L_{s,T}(X) = -(I_T + \alpha I_{T/2}) + \beta R_s + \gamma R_{ortho} \tag{5}$$

where $\alpha, \beta, \gamma$ are tradeoff weights and $R_{ortho} = ||\Sigma_1 - I_d||^2_{fro}$ is the orthonormality penalty discussed in Sec. 2.1, with $\Sigma_1 \in R^{d \times d}$ corresponding to the top left sub-matrix of $\Sigma_{2T}$ estimated from the latent sequence $Z = e(X \odot M)$. The whole framework of DAPC is illustrated in Figure 1.

## 2.3 A Probabilistic Interpretation of DAPC

We now discuss a probabilistic interpretation of DAPC in the Variational AutoEncoder (VAE) framework (Kingma & Welling, 2014). Let $X = (X_p, X_f)$ and $Z = (Z_p, Z_f)$, where the subscripts $p$ and $f$ denote *past* and *future* respectively. Consider a generative model, where the prior distribution is $p(Z_p, Z_f) \sim \mathcal{N}(\mathbf{0}, \Sigma)$. One can write down explicitly $p(Z_f|Z_p)$, which is a Gaussian with

$$\mu_{f|p} = \Sigma_{f,p} \Sigma^{-1}_{p,p} Z_p, \qquad \Sigma_{f|p} = \Sigma_{ff} - \Sigma_{f,p} \Sigma^{-1}_{p,p} \Sigma_{p,f}.$$

The linear dynamics in latent space are completely defined by the covariance matrix $\Sigma$. Large predictive information implies low conditional entropy $H(Z_f|Z_p)$.

Let $(Z_p, Z_f)$ generate $(X_p, X_f)$ with a stochastic decoder $g(X|Z)$. We only observe $X$ and would like to infer the latent $Z$ by maximizing the marginal likelihood of $X$. Taking a VAE approach, we parameterize a stochastic encoder $e(Z|X)$ for the approximate posterior, and derive a lower bound for the maximum likelihood objective. Different from standard VAE, here we would not want to parameterize the prior to be a simple Gaussian, in which case the $Z_p$ and $Z_f$ are independent and have zero mutual information. Instead we encourage the additional structure of high predictive information for the prior. This gives us an overall objective as follows:

$$\min_{\Sigma, e, g} \int \hat{p}(X) \left\{ \int -e(Z|X) \log g(X|Z) dz + KL(e(Z|X) \| p(Z)) \right\} dx - \eta I_T(\Sigma)$$

where $\hat{p}(X)$ is the empirical distribution over training data, the first term corresponds to the reconstruction loss, the second term measures the KL divergence between approximate posterior and the prior, and the last term is the PI defined in (2).

The challenge is how to parameterize the covariance $\Sigma$. We find that simply parameterizing it as a positive definite matrix, e.g., $\Sigma = AA^T$, does not work well in our experiments, presumably because there is too much flexibility with such a formulation. What we find to work better is the pseudo-input technique discussed in VampPrior (Tomczak & Welling, 2018): given a set of pseudo-sequences $X^*$ which are learnable parameters (initialized with real training sequences), we compute the sample covariance from $e(Z|X^*)$ as $\Sigma$.

This approach yields an overall objective very similar to (5), with the benefit of a well-defined generative model (and the Gaussian assumption being perfectly satisfied), which allows us to borrow learning/inference techniques developed in the VAE framework. For example, masking the input for the encoder can be seen as amortized inference regularization (Shu et al., 2018). We show experimental results on this probabilistic DAPC in Appendix B and C. In general, probabilistic DAPC performs similarly to the deterministic counterpart, though the training process is more time and memory intensive. On the other hand, these empirical results show that deviating from the Gaussian assumption, as is the case for deterministic DAPC, does not cause significant issues for representation learning in practice if proper regularization is applied.

Related to this interpretations are VAE-base sequential models (Chung et al., 2015; Hsu et al., 2017; Li & Mandt, 2018) that also use reconstruction and enforce different structures/dynamics in the latent space. Most of them are designed for the purpose of generating high quality sequence data, while the qualities of their latent representations are mostly not shown for downstream tasks.

## 3 RELATED WORK

Mutual information (MI) maximization is a principal approach for representation learning (Bell & Sejnowski, 1995), where the objective is to maximize the MI estimate between learnt representations and inputs. The currently dominant approach for estimating MI is based on contrastive learning. For sequence data, CPC (Oord et al., 2018) uses representations at current time as a classifier to discriminate inputs of nearby frames (positive samples) from inputs of far-away steps or inputs from other sequences (negative samples) with a cross-entropy loss; this leads to the noise-contrastive estimation (NCE, Gutmann & Hyvärinen, 2010). Deep InfoMax (DIM, (Hjelm et al., 2018)) generalizes the NCE estimator with a few variants, and proposes to maximize MI between global summary features and local features from intermediate layers (rather than the inputs as in CPC). SimCLR (Chen et al., 2020) extends the contrastive loss to use a nonlinear transformation of the representation (than the representation itself) as a classifier for measuring MI. Contrastive Multiview Coding (Tian et al., 2019) generalizes the contrastive learning frame to multiple views. Momentum Contrast (He et al., 2020) saves memory with a dynamic dictionary and momentum encoder.

Meanwhile, there have been concerns about the contrastive learning framework. One concern is that postive and negative sample selection is sometimes time and memory consuming. To address this issue, BYOL (Grill et al., 2020) proposes to get rid of negative samples by learning a target network in an online fashion and gradually bootstrapping the latent space. Another concern is regarding the MI estimation. Though contrastive learning has an MI backbone, Tschannen et al. (2019) suggests that the inductive bias of the feature extractor and parametrization of estimators might contribute more than the MI estimate itself. Ozair et al. (2019); McAllester & Stratos (2020) raise the concern

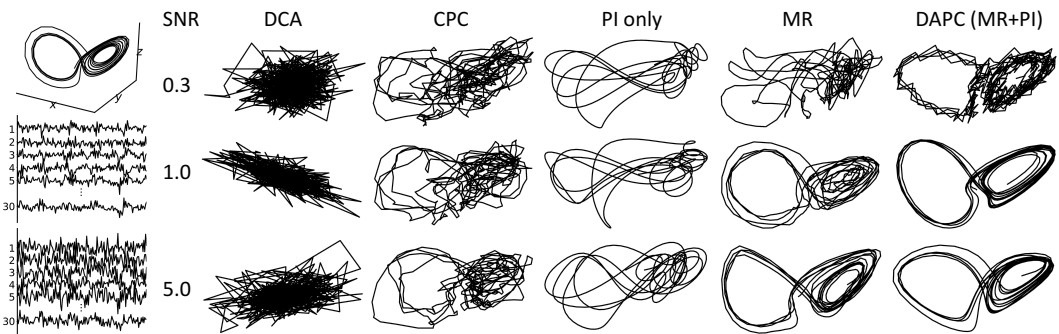

Figure 2: **Left panel**. Top: the ground-truth 3D Lorenz attractor. Middle: the 30D non-linearly lifted trajectory. Bottom: corrupted 30D trajectory by white noise with SNR=0.3. **Right Panel**. 3D trajectories extracted by different methods for three SNR levels: 0.3, 1.0, 5.0.

that the MI lower bound used by contrastive learning might be too loose, and propose to use an estimate based on Wasserstein distance.

Unlike prior work, our principle for sequence representation learning is to maximize the MI between past and future latent representations, rather than the MI between representations and inputs (or shallow features of inputs). Partially motivated by the above concerns, our mutual information estimate requires no sampling and is exact for Gaussian random variables. To keep useful information from input, we use a masked reconstruction loss which has been effective for sequence data (text and speech), with an intuition resembling that of denoising autoencoders (Vincent et al., 2010).

Note that by the data processing inequality, methods that maximize mutual information between current representation and future inputs also *implicitly* maximizes an upper bound of mutual information between high level representations, since $MI(Z^{past}, Z^{future}) \leq MI(Z^{past}, X^{future})$. Our method *explicitly* maximizes the mutual information between high level representations itself, while having another regularization term (masked reconstruction) that maximizes information between current input and current representations. Our results indicate that explicitly modeling the trade-off between the two can be advantageous.

In the audio domain where we will demonstrate the applicability of our method, there has been significant interest in representation learning for reducing the need for supervised data. Both contrastive learning based (Schneider et al., 2019; Baevski et al., 2019; Jiang et al., 2019) and reconstruction-based (Chorowski et al., 2019; Chung et al., 2019a; Song et al., 2019; Wang et al., 2020; Chung & Glass, 2020; Ling et al., 2020; Liu et al., 2020) methods have been studied, as well as methods that incorporate multiple tasks (Pascual et al., 2019a; Ravanelli et al., 2020). Our work promotes the use of a different MI estimate and combines different intuitions synergistically.

## 4 EXPERIMENTS

### 4.1 NOISY LORENZ ATTRACTOR

Lorenz attractor ("butterfly effect", see Appendix B) is a 3D time series depicting a chaotic system (Pchelintsev, 2014), as visualized in Figure 2. We design a challenging dimension reduction task for recovering the Lorenz attractor from high dimensional noisy measurements. We first lift the 3D clean signals to 30D with a neural network of 2 hidden layers, each with 128 *elu* units (Clevert et al., 2015). This lifting network has weights and biases drawn randomly from $\mathcal{N}(0, 0.2)$. In addition, we corrupt the 30D lifted signals with white noise to obtain three different signal-to-noise ratio (SNR) levels, 0.3, 1.0, and 5.0, and use the noisy 30D measurements (see bottom left of Figure 2) as input for representation learning methods to recover the true 3D dynamics.

We generate a long Lorenz attractor trajectory using the governing differential equations, and chunk the trajectory into segments of 500 time steps. We use 250, 25 and 25 segments for training, validation, and test splits respectively. After the model selection on the validation split based on the $R^2$ regression score which measures the similarity between recovered and ground truth trajectories,

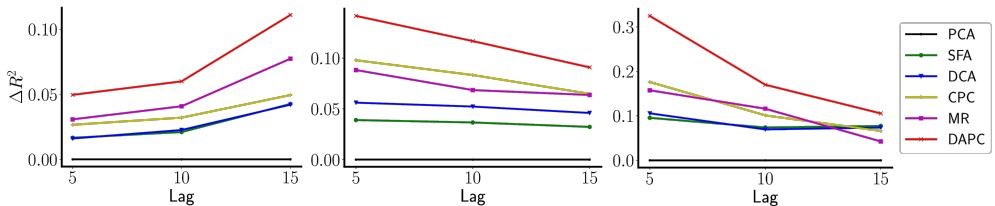

Figure 3: Different models' $R^2$ score improvements over PCA for three forecasting tasks, each with three different $lag$ values. Left: temperature. Middle: dorsal hippocampus. Right: motor cortex.

Table 1: The $R^2$ scores for full reconstruction (FR), full reconstruction with PI (FR+PI), masked reconstruction (MR) and DAPC (MR+PI). We also demonstrate the improvements in percentage brought by adding PI. On average, PI improves full reconstruction by 4.09% and masked reconstruction by 17.22%. Additionally, in the last column, we show PI objective alone is not powerful enough to learn predictive components ($R^2$ scores are low).

| dataset | lag | FR | FR+PI | Improvement | MR | MR+PI | Improvement | PI only |
|---|---|---|---|---|---|---|---|---|
| Temp | 5 | 0.721 | 0.725 | 0.58% | 0.705 | 0.724 | 2.68% | 0.698 |
| | 10 | 0.672 | 0.685 | 1.99% | 0.672 | 0.691 | 2.85% | 0.645 |
| | 15 | 0.675 | 0.686 | 1.64% | 0.673 | 0.707 | 4.99% | 0.632 |
| HC | 5 | 0.252 | 0.260 | 3.24% | 0.251 | 0.304 | 21.25% | 0.169 |
| | 10 | 0.222 | 0.231 | 4.46% | 0.222 | 0.271 | 21.72% | 0.137 |
| | 15 | 0.183 | 0.194 | 5.97% | 0.205 | 0.232 | 13.28% | 0.085 |
| M1 | 5 | 0.390 | 0.405 | 4.03% | 0.352 | 0.519 | 47.45% | 0.239 |
| | 10 | 0.372 | 0.394 | 5.83% | 0.369 | 0.422 | 14.64% | 0.156 |
| | 15 | 0.268 | 0.293 | 9.10% | 0.241 | 0.304 | 26.10% | 0.076 |

our model is applied to the test split. The optimal model uses $T = 4$, $s = 0$, $\alpha = 0$, $\gamma = 0.1$, and $\beta = 0.1$ is set to balance the importance of PI and the reconstruction error.

We compare DAPC to representative unsupervised methods including DCA (Clark et al., 2019), CPC (Oord et al., 2018), pure PI learning which corresponds to DAPC with $\beta = 0$, and masked reconstruction (MR, Wang et al., 2020) which corresponds to DAPC without the PI term. Except for DCA which corresponds to maximizing PI with a linear feedforward network, the other methods use bidirectional GRUs (Chung et al., 2014) for mapping the inputs into feature space (although uni-GRU performs similarly well). A feedforward DNN is used for reconstruction in MR and DAPC.

We show the latent representations of different methods in Figure 2 (right panel). DCA fails completely since its encoder network has limited capacity to invert the nonlinear lifting process. CPC is able to recover the 2 lobes, but the recovered trajectory is chaotic. Maximizing PI alone largely ignores the global structure of the data. MR is able to produce smoother dynamics for high SNRs, but its performance degrades quickly in the noisier scenarios. DAPC recovers a latent representation which has overall similar shapes to the ground truth 3D Lorenz attractor, and exhibits smooth dynamics enforced by the PI term. In Appendix B, we provide the $R^2$ scores for different methods. These results quantitatively demonstrate the advantage of DAPC across different noise levels.

## 4.2 Forecasting with Linear Regression

We then demonstrate the predictive power of learnt representations in downstream forecasting tasks on 3 real-world datasets used by Clark et al. (2019), involving multi-city temperature time series data (Temp, Beniaguev (2017)), dorsal hippocampus study (HC, Glaser et al. (2020)), and motor cortex (M1, O'Doherty et al. (2018)). For each model, unsupervised representation learning is performed on the training set with a uni-directional GRU, which prevents information leakage from the future. After that, we freeze the model and use it as a feature extractor. The representations at each time step are used as inputs for predicting the target at a future time step. As an example, we can extract a representation for today's weather based on past weather only (as the encoder is uni-directional), and use it to predict future temperature which is $lag$ days away (a larger $lag$ generally leads to a more difficult forecasting task). Following Clark et al. (2019), the predictor from the extracted

feature space to the target is a linear mapping, trained on samples of paired current feature and future target, using a least squares loss. We use the same feature dimensionality as in their work, for each dataset. These forecasting tasks are evaluated by the $R^2$ regression score, which measures the linear predictability. More details can be found in Appendix C.

Besides DCA, CPC and MR, we further include PCA and SFA (Wiskott & Sejnowski, 2002) (similar to DCA with $T = 1$ for PI estimation), which are commonly used linear dimension reduction methods in these fields. PCA serves as the baseline and we report $R^2$ score improvements from other methods. Figure 3 gives the performances of different methods for the three datasets, with three different $lag$s (the number of time steps between current and future for the forecasting task): 5, 10, and 15. DAPC consistently outperforms the other methods. In Table 1, we show how PI helps DAPC improve over either full reconstruction (e.g., classical auto-encoder) or masked reconstruction, and how reconstruction losses help DAPC improve over PI alone on this task. These results demonstrate that the two types of losses, or the two types of mutual information (MI between input and latent, and MI between past and future) can be complementary to each other.

### 4.3 Pretraining for Automatic Speech Recognition (ASR)

A prominent usage of representation learning in speech processing is to pretrain the acoustic model with an unsupervised objective, so that the resulting network parameters serve as a good initialization for the supervised training phase using labeled data (Hinton et al., 2012). As supervised ASR techniques have been improved significantly over the years, recent works start to focus on pre-training with large amounts of unlabeled audio data, followed by finetuning on much smaller amounts of supervised data, so as to reduce the cost of human annotation.

We demonstrate different methods on two commonly used speech corpora for this setup: Wall Street Journal (Paul & Baker, 1992) and LibriSpeech (Panayotov et al., 2015). For WSJ, we pretrain on *si284* partition (81 hours), and finetune on *si84* partition (15 hours) or the *si284* partition itself. For Librispeech, we pretrain on the *train_960* partition (960 hours) and finetune on the *train_clean_100* partition (100 hours). Standard dev and test splits for each corpus are used for validation and testing.

In the experiments, we largely adopt the transformers-based recipe from ESPnet (Watanabe et al., 2018), as detailed in Karita et al. (2019), for supervised finetuning. We provide the details regarding model architecture and data augmentation in Appendix D. Note that we have spent effort in building strong ASR systems, so that our baseline (without pretraining) already achieves low WERs and improving over it is non-trivial. This can be seen from the result table where our baseline is often stronger than the best performance from other works. In the pretraining stage, we pretrain an encoder of 14 transformer layers, which will be used to initialize the first 14 layers of ASR model. For masked reconstruction, we use 2 frequency masks as in finetuning, but found more time masks can improve pretraining performance. We set the number of time masks to 4 for WSJ, and 8 for LibriSpeech which has longer utterances on average.

The hyperparameters we tune include $T, s, \alpha, \beta$ and $\gamma$ from our learning objective. We select hyperparameters which give the best dev set WER, and report the corresponding test set WER. In the end, we use $T = 4$ for estimating the PI term, $\gamma = 0.05$, $\beta = 0.005$ and set $s = 2$ for WSJ and $s = 1$ for LibriSpeech if we use shifted reconstruction. Since the pretraining objective is a proxy for extracting the structure of data and not fully aligned with supervised learning, we also tune the number of pretraining epochs, which is set to 5 for WSJ and 1 for LibriSpeech. Other parameters will be shown in the ablation studies presented in Appendix D.

We perform an ablation study for the effect of different variants of DAPC (MR+PI) on the WSJ dataset, and give dev/test WERs in Table 2. We tune the hyperparameters for multi-scale PI and shifted reconstruction for the 15-hour finetuning setup, and observe that each technique can lead to further improvement over the basic DAPC, while combining them delivers the best performance. The same hyperparameters are used for the 81-hour finetuning setup, and we find that with more supervised training data, the baseline without pretraining obtains much lower WERs, and pure masked reconstruction only slightly improves over the baseline, while the strongest DAPC variant still achieves 7.9% and 11.7% relative improvements on *dev93* and *eval92* respectively.

In Table 3, we provide a more thorough comparison with other representation learning methods on the LibriSpeech dataset. We compare with CPC-type mutual information learning methods including

Table 2: Ablation study on different variants of DAPC. We give WERs (%) of ASR models pretrained with different variants on WSJ. Models are pretrained on 81 hours, and finetuned on either 15 hours or 81 hours. All results are averaged over 3 random seeds.

| Methods | dev93 | eval92 |
|---|---|---|
| Finetune on 15 hours | | |
| w.o. pretrain | 12.91±0.36 | 8.98±0.44 |
| PI only | 12.54±0.32 | 9.02±0.43 |
| MR | 12.27±0.23 | 8.15±0.34 |
| DAPC | 12.31±0.36 | 7.74±0.20 |
| DAPC + multi-scale PI | 12.15±0.35 | 7.64±0.15 |
| DAPC + shifted recon | 11.93±0.16 | 7.68±0.05 |
| DAPC + both | **11.57**±0.22 | **7.34**±0.13 |
| Finetune on 81 hours | | |
| w.o. pretrain | 6.34±0.13 | 3.94±0.33 |
| MR | 6.24±0.14 | 3.84±0.09 |
| DAPC | 5.90±0.16 | 3.58±0.08 |
| DAPC + both | **5.84**±0.04 | **3.48**±0.08 |

Table 3: WERs (%) obtained by ASR models pretrained with different representation learning methods on the *test_clean* partition of Librispeech. Models are pretrained on 960h unlabeled data and finetuned on 100h labeled data. Our results are averaged over 3 random seeds.

| Methods | WER (%) |
|---|---|
| wav2vec (Schneider et al., 2019) | 6.92 |
| discrete BERT+vq-wav2vec (Baevski et al., 2019) | 4.5 |
| wav2vec 2.0 (Baevski et al., 2020) | **2.3** |
| DeCoAR (Ling et al., 2020) | 6.10 |
| TERA-large (Liu et al., 2020) | 5.80 |
| MPE (Liu & Huang, 2020) | 9.68 |
| Bidir CPC (Kawakami et al., 2020) | 8.70 |
| w.o. pretrain | 5.11±0.20 |
| MR | 5.02±0.09 |
| DAPC | 4.86±0.08 |
| DAPC+multi-scale PI+shifted recon | **4.70**±0.02 |

wav2vec (Schneider et al., 2019), vq-wav2vec which performs CPC-type learning with discrete tokens followed by BERT-style learning (Baevski et al., 2019), and wav2vec 2.0 which incorporates masking into contrastive learning (Baevski et al., 2020). We also compare with two reconstruction-type learning approaches DeCoAR (Ling et al., 2020) and TERA (Liu et al., 2020); comparisons with more methods are given in Appendix D. Observe that DAPC and its variant achieve lower WER than MR: though our baseline is strong, DAPC still reduces WER by 8%, while MR only improves by 1.76%. This shows the benefit of PI-based learning in addition to masked reconstruction. Our method does not yet outperform vq-wav2vec and wav2vec 2.0; we suspect it is partly because our models have much smaller sizes (around 30M weight parameters) than theirs (vq-wav2vec has 150M weight parameters, and wav2vec has 300M weight parameters for the acoustic model, along with a very large neural language model) and it is future work to scale up our method. In Appendix D, we provide additional experimental results where the acoustic model targets are subwords. DAPC achieves 15% relative improvement over the baseline (without pretraining), showing that our method is generally effective for different types of ASR systems.

## 5 CONCLUSIONS

In this work, we have proposed a novel representation learning method, DAPC, for sequence data. Our learnt latent features capture the essential dynamics of the underlying data, contain rich information of both input observations and context states, and are shown to be useful in a variety of tasks. As future work, we may investigate other predictive information estimators that further alleviate the Gaussian assumption. On the other hand, more advanced variational inference techniques may be applied to the probabilistic version of DAPC to boost the performance. DAPC provides a general alternative for mutual information-based learning of sequence data and we may investigate its potential usage in other domains such as NLP, biology, physics, etc.

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

## A  DERIVATION OF PREDICTIVE INFORMATION

In this section, we give a self-contained derivation of the predictive information.

A multivariate Gaussian random variable $X \in \mathbb{R}^N$ has the following PDF:

$$p(x) = \frac{1}{\sqrt{2\pi}^N \sqrt{|\Sigma|}} \exp\left(-\frac{1}{2}(x-\mu)^T \Sigma^{-1}(x-\mu)\right)$$

where $\mu$ is the mean and $\Sigma$ is the covariance matrix with $\Sigma = \mathbb{E}[(X - \mathbb{E}[X])(X - \mathbb{E}[X])]$.

From the definition of entropy

$$H(X) = -\int p(x) \ln p(x) dx$$

we can derive the entropy formula for multivariate Gaussian

$$
\begin{aligned}
H(X) &= -\int p(x) \ln \frac{1}{\sqrt{2\pi}^N \sqrt{|\Sigma|}} \exp(-\frac{1}{2}(x-\mu)^T \Sigma^{-1}(x-\mu)) dx \\
&= -\int p(x) \ln \left(\frac{1}{(\sqrt{2\pi})^N \sqrt{|\Sigma|}}\right) dx - \int p(x) \left(-\frac{1}{2}(x-\mu)^T \Sigma^{-1}(x-\mu))\right) dx \\
&= \frac{1}{2} \ln((2\pi e)^N |\Sigma|).
\end{aligned}
\tag{6}
$$

Consider the joint Gaussian distribution $\begin{pmatrix} X \\ Y \end{pmatrix} \sim \mathcal{N}(\mu, \Sigma) = \mathcal{N}\left(\begin{pmatrix} \mu_X \\ \mu_Y \end{pmatrix}, \begin{pmatrix} \Sigma_{XX} & \Sigma_{XY} \\ \Sigma_{YX} & \Sigma_{YY} \end{pmatrix}\right)$
where $X \in \mathbb{R}^p$, and $Y \in \mathbb{R}^q$. We can plug in the entropy in (6) and obtained

$$
\begin{aligned}
MI(X,Y) &= H(X) + H(Y) - H(X,Y) \\
&= \frac{1}{2} \ln((2\pi e)^p |\Sigma_{XX}|) + \frac{1}{2} \ln((2\pi e)^q |\Sigma_{YY}|) - \frac{1}{2} \ln((2\pi e)^{p+q} |\Sigma|) \\
&= -\frac{1}{2} \ln \frac{|\Sigma|}{|\Sigma_{XX}||\Sigma_{YY}|}.
\end{aligned}
$$

For a latent sequence $Z = \{z_1, z_2, ...\}$ where $z_i \in \mathbb{R}^d$, we define $Z_t^{past} = \{z_{t-T+1}, ..., z_t\}$, and $Z_t^{future} = \{z_{t+1}, ..., z_{t+T}\}$. Based on our stationarity assumption, all the length-$2T$ windows of states within the range are drawn from the same Gaussian distribution with covariance $\Sigma_{2T}(Z)$, and similarly for all the length-$T$ windows. As a result and under the stationary assumption, $H(Z_t^{past}) = H(Z_t^{future}) = \frac{1}{2} \ln((2\pi e)^{Td} |\Sigma_T(Z)|)$, $H(Z_t^{past}, Z_t^{future}) = \frac{1}{2} \ln((2\pi e)^{2Td} |\Sigma_{2T}(Z)|)$, and

$$
\begin{aligned}
I_T &= MI(Z_t^{past}, Z_t^{future}) \\
&= H(Z_t^{past}) + H(Z_t^{future}) - H(Z_t^{past}, Z_t^{future}) \\
&= \frac{1}{2} \ln((2\pi e)^{Td} |\Sigma_T(Z)|) + \frac{1}{2} \ln((2\pi e)^{Td} |\Sigma_T(Z)|) - \frac{1}{2} \ln((2\pi e)^{2Td} |\Sigma_{2T}(Z)|) \\
&= \ln |\Sigma_T(Z)| - \frac{1}{2} \ln |\Sigma_{2T}(Z)|
\end{aligned}
$$

The predictive information $I_T$ only depend on $T$ but not specific time index $t$.

## B  LORENZ ATTRACTOR

The Lorenz attractor system (also called "butterfly effect") is generated by the following differential equations: (Strogatz, 2018; Clark et al., 2019):

$$
\begin{aligned}
dx &= \sigma(y - x) \\
dy &= x(\rho - z) - y \\
dz &= xy - \beta z
\end{aligned}
\tag{7}
$$

where $(x, y, z)$ are the 3D coordinates, and we use $\sigma = 10, \beta = 8/3, \rho = 28$. The integration step for solving the system of equations is $5 \times 10^{-3}$.

We lift the 3D trajectory into 30D using a nonlinear neural network with 2 hidden layers, each with 128 neurons and the Exponential Linear Unit (Clevert et al., 2015). We further perturb the 30D trajectory with additive Gaussian noise to obtain datasets of three different Signal-to-Noise Ratios (SNRs, ratio between the power of signal and the power of noise): 0.3, 1.0, and 5.0. The smaller the SNR is, the more challenging it is to recover the clean 3D trajectory (see the left panel of Figure 2 for the comparison between noisy 30D trajectory and the clean 30D trajectory).

Table 4: The $R^2$ scores of recovered 3D trajectory of noisy Lorenz attractor by different methods.

| SNR | DCA | CPC | PI | MR | DAPC-det | DAPC-prob |
|-----|-----|-----|-----|-----|----------|-----------|
| 0.3 | 0.084 | 0.676 | 0.585 | 0.574 | 0.865 | 0.816 |
| 1.0 | 0.153 | 0.738 | 0.597 | 0.885 | 0.937 | 0.943 |
| 5.0 | 0.252 | 0.815 | 0.692 | 0.929 | 0.949 | 0.949 |

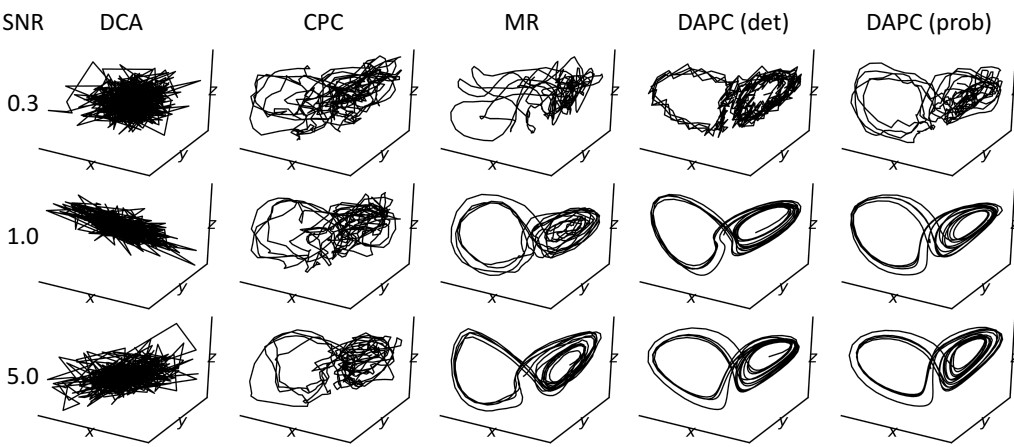

Figure 4: Recovery of 3D trajectory of noisy Lorenz attractor by different methods.

We compare both deterministic and probabilistic DAPC against representative unsupervised methods including DCA (Clark et al., 2019), CPC (Oord et al., 2018), pure PI learning which corresponds to DAPC with $\beta = 0$, and masked reconstruction (MR) (Wang et al., 2020). Except for DCA which corresponds to maximizing PI with a linear orthogonal feedforward net, the other methods use bidirectional GRU (Chung et al., 2014) for mapping the inputs into feature space (although uni-GRU performs similarly well). A feedforward DNN is used for reconstruction in MR and DAPC.

More specifically, CPC, MR, DAPC all use the bidirectional GRU where the learning rate is 0.001, and dropout rate is 0.7. Our GRU has 4 encoding layers with hidden size 256. The batch size is (20, 500, 30). For CPC, the temporal lag k=4. For DAPC, $\beta = 0.1$, $T = 4$, $s = 0$, $\alpha = 0$, $\gamma = 0.1$. For masked reconstruction, we use at most 2 masks on the frequency axis with width up to 5, and at most 2 masks on the time axis with width up to 40. The DNN decoder has 3 hidden layers, each with size 512. DCA's setup is completely adopted from Clark et al. (2019). The same architectures are used in the forecasting tasks in Appendix C.

Figure 4 provides qualitatively results for the recovered 3D trajectories by different methods (Figure 2 in the main text contains a subset of the results shown here). Observe that DCA fails in this scenario since its feature extraction network has limited capacity to invert the nonlinear lifting process. CPC is able to recover the 2 lobes, but the recovered signals are chaotic. Masked reconstruction is able to produce smoother dynamics for high SNRs, but its performance degrades quickly in the more noisy scenarios. Both deterministic and probabilistic DAPC recover a latent representation which has overall similar shapes to the ground truth 3D Lorenz attractor, and exhibits smooth dynamics enforced by the PI term.

We quantitatively measure the recovery performance with the $R^2$ score, which is defined as coefficient of determination. $R^2$ score normally ranges from 0 to 1 where 1 means the perfect fit.

Negative scores indicate that the model fits the data worse than a horizontal hyperplane. The $R^2$ results are given in Table 4. Our results quantitatively demonstrate the clear advantage of DAPC across different noise levels.

Table 5: The $R^2$ scores for the ablation study of (deterministic) DAPC for Lorenz attractor.

| SNR | Full Recon | uni-GRU | Regular |
|---|---|---|---|
| 0.3 | 0.803 | 0.857 | 0.865 |
| 1.0 | 0.812 | 0.905 | 0.937 |
| 5.0 | 0.852 | 0.903 | 0.949 |

Table 6: The $R^2$ scores for full reconstruction only and full reconstruction with PI.

| SNR | Full Recon only | Full Recon with PI |
|---|---|---|
| 0.3 | 0.441 | 0.803 |
| 1.0 | 0.737 | 0.812 |
| 5.0 | 0.802 | 0.852 |

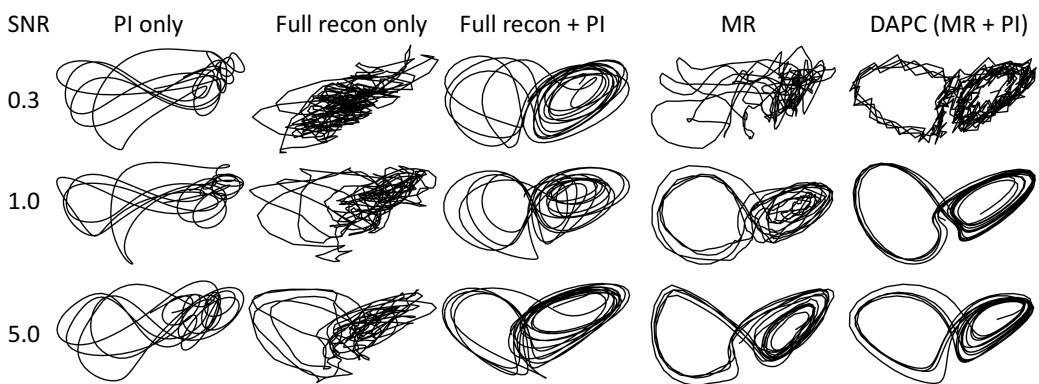

Figure 5: Illustration of how PI can improve both full reconstruction and masked reconstruction (MR). We can observe that PI can greatly improve the recovery quality, especially when SNR is low (very noisy).

Table 7: The $R^2$ scores for CPC with different temporal lags (k).

| SNR | k=2 | k=4 | k=6 | k=8 | k=10 |
|---|---|---|---|---|---|
| 0.3 | 0.477 | 0.676 | 0.663 | 0.658 | 0.608 |
| 1.0 | 0.556 | 0.738 | 0.771 | 0.721 | 0.642 |
| 5.0 | 0.652 | 0.815 | 0.795 | 0.775 | 0.717 |

We also give an ablation study on several components of DAPC in Table 5, where we attempt full reconstruction without masking, masked reconstruction with unidirectional encoder uni-GRU, and the regular setup (masked reconstruction + bi-GRU). Using full reconstruction yields worse results than using masked reconstruction at all noise levels, while uni-GRU degrades the performance less.

We show in Table 6 and Figure 5 how PI can improve both full reconstruction and masked reconstruction. In Table 6, when SNR=0.3, PI can greatly boost the performance of full reconstruction. We also tuned the temporal lag parameter k w.r.t. both quantitative and qualitative results (Table 7 and Figure 6). CPC performance starts to deteriorate after k=8, while k=4, 6, 8 have similar results. Based on the $R^2$ scores, we select k=4 as our final temporal lag. Similar tuning is also performed for CPC on the downstream forecasting experiments in Appendix C.

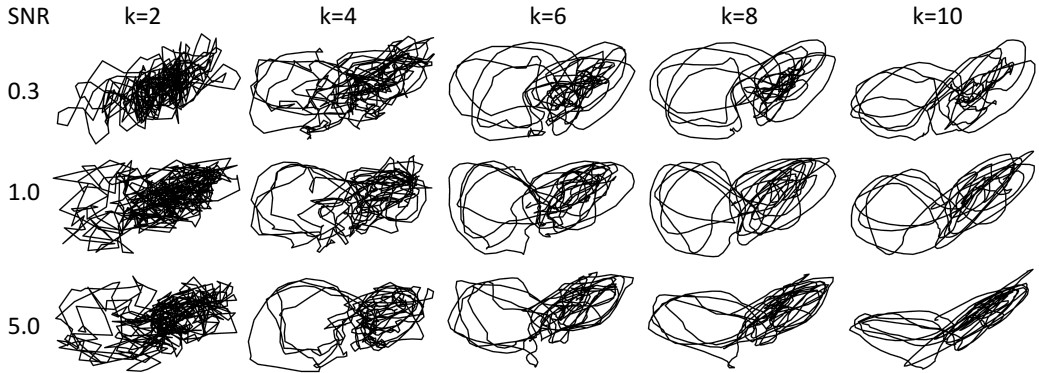

Figure 6: Qualitative recovery results by CPC w.r.t. different temporal lags (k).

## C  DOWNSTREAM FORECASTING WITH LINEAR REGRESSION

For each of the three datasets used in downstream forecasting tasks, we divide the original dataset into training/validation/testing splits. Unsupervised representation learning is performed on the training split, validated on the validation split and learns an encoder $e(\cdot)$.

Denote the test sequence $X = \{x_1, x_2, ..., x_L\}$. The learnt $e(\cdot)$ transforms $X$ into a sequence $Z = \{z_1, z_2, ..., x_L\}$ of the same length. Note that we use uni-directional GRU for representation learning so that and no future information is leaked. For the forecasting tasks, $z_i$ will be used to predict a target $y_i$ which corresponds to an event of interest. For the multi-city temperature dataset (Beniaguev, 2017), $y_i$ represents future multi-city temperatures, i.e., $y_i = x_{i+lag}$ with $lag > 0$. For the hippocampus study (Glaser et al., 2020), $x_i$ is the multi-neuronal spiking activity of 55 single units recorded in rat hippocampal CA1 and $y_i$ is a future location of the rat. In the motor cortex dataset (O'Doherty et al., 2018), $x_i$'s are collected from multi-neuronal spiking activity of 109 single units recorded in monkey primary motor cortex (M1), and $y_i$'s are future behavior variables such as cursor kinematics. The problems tend to be more challenging with larger $lag$.

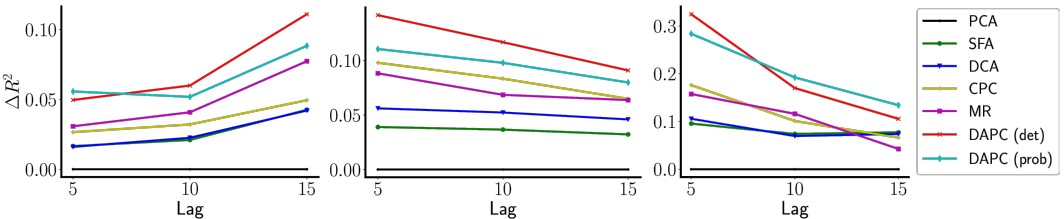

Figure 7: Different models' $R^2$ score improvements over PCA: **Left)** temperature, **Middle)** dorsal hippocampus and **Right)** motor cortex datasets. Lag is the number of time steps the future event is ahead of the current latent state.

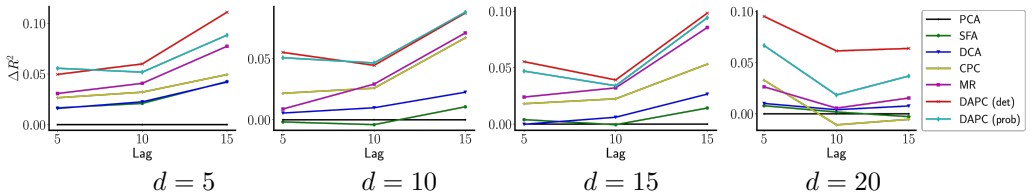

Figure 8: On temperature dataset, we analyze the performances of different latent dimensions (dimension of $z_i$): 5, 10, 15, 20. $\Delta R^2$ corresponds to $R^2$ score improvement over PCA.

The performance for forecasting tasks is measured by the linear predictability from $z_i$ to $y_i$. Specifically, we solve the linear regression problem with inputs being the $(z_i, y_i)$ pairs, and measure the

$R^2$ score between the prediction and ground-truth target. We use the $R^2$ score from the PCA projection as a baseline, and provide the improvements over PCA obtained by different representation learning methods in Figure 7 (so PCA's $\Delta R^2$ is 0). Both deterministic and probabilistic DAPC consistently outperform other methods across all three datasets, with the deterministic version slightly outperforming the probabilistic one. Additionally, we provide sensitivity study of the latent dimensionality for all methods in Figure 8, and DAPC outperforms others consistently across different dimensionalities. Table 8 shows the temporal lag tuning for CPC on the temperature dataset.

Table 8: The $R^2$ scores for CPC with different temporal lags (k) on the temperature dataset.

| d  | k=2   | k=4   | k=6   | k=8   | k=10  |
|----|-------|-------|-------|-------|-------|
| 5  | 0.685 | 0.701 | 0.690 | 0.687 | 0.681 |
| 10 | 0.661 | 0.663 | 0.675 | 0.631 | 0.622 |
| 15 | 0.633 | 0.645 | 0.637 | 0.634 | 0.617 |

## D  ADDITIONAL RESULTS FOR AUTOMATIC SPEECH RECOGNITION (ASR)

The acoustic model is trained with a multi-task objective (Watanabe et al., 2017) which combines attention Chorowski et al. (2015); Chan et al. (2016) and CTC (Graves et al., 2006) losses, for predicting the output character sequence. We extract 80D fbank features plus 3D pitch features from audio, with a frame size of 25ms and hop size of 10ms. Every 3 consecutive frames are stacked to obtain the input sequence for the acoustic model. During ASR finetuning, the encoder shared by both attention and CTC consists of 14 transformer layers for WSJ and 16 layers for LibriSpeech, while the decoder consists of 6 transformer layers. All attention operations use 4 heads of 64 dimensions each, and the output of multi-head attention goes through a one-hidden-layer position-wise feed-forward network of 2048 ReLU units, before it is fed into the next layer. During finetuning, we apply SpecAugment (Park et al., 2019) to reduce overfitting, with *max_time_warp* set to 5 (frames), two frequency masks of width up to 30 frequency bins, and two time masks of width up to 40 frames. We use the Adam optimizer with a warmup schedule for the learning rate. Weight parameters of the last 10 finetuning epochs is averaged to obtain the final model. For word-level decoding, we use a word RNNLM trained on the language model training data of each corpus, with a vocabulary size of 65K for WSJ, and 200K for LibriSpeech. We use the lookahead scores derived from word RNNLM during beam search, for selecting promising character tokens at each step, as done by Hori et al. (2018). A beam size of 20 is used for decoding.

In Table 9, we provide a thorough comparison with other representation learning methods on the LibriSpeech dataset. We compare with CPC-type mutual information learning methods including wav2vec (Schneider et al., 2019), vq-wav2vec which performs CPC-type learning with discrete tokens followed by BERT-stype learning (Baevski et al., 2019), and a more recent extension wav2vec 2.0 (Baevski et al., 2020). We also compare with two reconstruction-type learning approaches De-CoAR (Ling et al., 2020) and TERA (Liu et al., 2020). Note that TERA is quite similar to MR (Wang et al., 2020) in performing masked reconstruction, although it adds recurrent layers to transformer-learnt representations for finetuning, while our implementation of MR uses a pure transformer-based architecture throughout. We believe the advantage of MR over TERA mainly comes from the acoustic model (attention vs. CTC) and a stronger language model (RNNLM vs. n-gram). Observe that DAPC and its variant achieve lower WER than MR: though our baseline is strong, DAPC still reduces WER by 8%, while MR only improves by 1.76%. This shows the benefit of PI-based learning in addition to masked reconstruction. Compared to Table 3, Table 9 includes more recent works that are related to our DAPC. Futhermore, to show our improvement is robust to details of ASR recipe, we include the comparison between baseline (without pretraining), MR, and DAPC when the ASR recipe uses 5000 unigrams as token set and decodes with token-level RNNLM. These results are denoted with "(sub-word)" in Table 9. The relative merits between methods are consistent with those observed for the character recipe. DAPC obtains a relative improvement of 15% over the baseline on *test_clean* ($6.81\% \rightarrow 5.79\%$).

Furthermore, we compare DAPC with other state-of-the-art methods on WSJ in Table 10. Comparisons among different methods based on their key features are shown in Table 11.

Table 9: WER results of different methods on LibriSpeech. All representation methods are pre-trained on the full corpus (960h) and finetuned on *train_clean_100* (100h). For MR and DAPC, the default ASR recipe uses characters as token set and decodes with word RNNLM. For results denoted with "(sub-word)", the ASR recipe uses 5000 unigrams as token set and decodes with token-level RNNLM. All the results are averaged over 3 seeds.

| Methods | *dev_clean* | *test_clean* |
|---|---|---|
| wav2vec (Schneider et al., 2019) | - | 6.92 |
| discrete BERT+vq-wav2vec (Baevski et al., 2019) | 4.0 | 4.5 |
| wav2vec 2.0 Baevski et al. (2020) | **2.1** | **2.3** |
| DeCoAR (Ling et al., 2020) | - | 6.10 |
| TERA-large (Liu et al., 2020) | - | 5.80 |
| MPE (Liu & Huang, 2020) | 8.12 | 9.68 |
| Bidir CPC (Kawakami et al., 2020) | 8.86 | 8.70 |
| MR (Wang et al., 2020) | 4.66 | 5.02 |
| MR (sub-word) | 5.57 | 6.18 |
| w.o. pretrain | 4.84 | 5.11 |
| DAPC | 4.52 | 4.86 |
| DAPC+multi-scale PI | 4.46 | 4.77 |
| DAPC+shifted recon | 4.50 | 4.80 |
| DAPC+multi-scale PI+shifted recon | **4.42** | **4.70** |
| w.o. pretrain (sub-word) | 6.16 | 6.81 |
| DAPC (sub-word) | 5.50 | 5.79 |

Table 10: WERs of models pretrained on 81h split *si284* and finetuned on 81h split *si284*.

| Methods | *dev93* WER (%) | *eval92* WER (%) |
|---|---|---|
| DeCoAR (Ling et al., 2020) | 8.34 | 4.64 |
| MPE (Liu & Huang, 2020) | 6.79 | 4.26 |
| MR (Wang et al., 2020) | 6.24 | 3.84 |
| w. o. pretrain (Attention+CTC) | 6.34 | 3.94 |
| DAPC + both | 5.84 | 3.48 |

Table 11: We compare different methods based on their key features: generative/discriminative, contrastive/non-contrastive, whether using past-future mutual information estimation (p-f MI), and masking.

| | generative | contrastive | p-f MI | masking |
|---|---|---|---|---|
| SFA | n | n | y | n |
| DCA | n | n | y | n |
| wav2vec | n | y | n | n |
| vq-wav2vec | n | y | n | y |
| DeCoAR | n | n | n | y |
| TERA | n | n | n | y |
| MPE | n | n | n | y |
| bidir CPC | n | y | n | n |
| DAPC | y | n | y | y |

