# OpenReview forum: "Representation Learning for Sequence Data with Deep Autoencoding Predictive Components"
_ICLR.cc/2021/Conference — ICLR 2021 Poster_

### Official Review · AnonReviewer4 · 2020-10-26
**Interesting approach, lacking some depth in the interpretation and experiments**

**Rating:** 5
**Confidence:** 3

**Review:**

This paper proposes Deep Autoencoding Predictive Components (DAPC), a self-supervised representation learning approach for sequential data. In this approach, the model learns to maximize the predictive information, which is the mutual information between past and future time windows. In order to avoid degenerate solutions, the proposed approach relies on a second loss that optimizes masked reconstructions.

Strengths:
* The paper is written clearly and is easy to follow.
* The proposed method seems to outperform the baselines across all considered tasks.
* The usage of predictive information as an objective on the latent representations is well motivated and seems to improve performance over a non-regularized autoencoder.

Weaknesses:
* The paper focuses exclusively on one interpretation of their approach, in which the optimization of the predictive information is the main objective and the reconstruction loss merely pushes the model to avoid degenerate solutions. Explicitly considering the reverse interpretation, in which the predictive information is a regularizer applied to an autoencoder to enforce a better latent structure, could provide more depth to the paper:
    1) Following this interpretation, it would make sense to include more references in the related works section describing autoencoding approaches for representation learning.
    2) Based on this interpretation, it would also be interesting to investigate the effect that the predictive information regularizer has on the autoencoding model. This is partially done in section 4.3, but missing from the other two experiments.
    3) Based on this interpretation, the probabilistic interpretation of DAPC (sec 2.3) can be reduced to describing a standard VAE with additional regularization on the latent space in the form of predictive information.

* The experiments show strong results for the proposed DAPC. However, it remains unclear how much fine-tuning has been done for the DAPC, especially in comparison to the baselines. For example, I would imagine that the CPC model might perform better on the lorenz attractor (Fig. 2) when optimizing the parameter k (which influences the temporal lag between the positive samples). The experiments would provide a much stronger point for the superiority of DAPC if there was a clear outline of the considered hyperparameters for all models. Additionally, details, such as the employed model architectures, that would be needed for reproducing the results are missing.

Additional comment:
In the related work section, the paper states “Unlike prior work, our principle for sequence representation learning is to maximize the MI between the past and future latent representations, rather than the MI between representations and inputs.”. That characterization of previous work is not entirely correct: CPC, for example, optimizes the MI $I(x_{t+k}, c_t)$ between the future input $x_{t+k}$ and the current representation $c_t$. Through the data processing inequality, this will result in the model implicitly also optimizing the MI $I(c_{t+k}, c_t)$ between the future and current representation (Tschannen et al., 2019).

---

> ### Author Response · Authors · 2020-11-20
> **we agree with your interpretation and we did tune k in CPC**
>
> * We agree both interpretations make sense and we just picked one of them to make our paper more clear.
>     **a)** We added some discussion in section 2 regarding other sequential VAE based models. But most of them are designed for the generation purpose instead of representation learning for downstream tasks. Regarding ASR, we have discussed reconstruction-based methods like APC, denoising autoencoders, masked reconstruction in section 2.2 and section 3.
>     **b)** We have added Table 5, Figure 5, Table 7 to illustrate the regularization effects of PI on full reconstruction. The results show that PI improves either full reconstruction or masked reconstruction for representation learning.
>     **c)** We agree with your point of probabilistic DAPC as VAE with additional regularization on the latent space (and the deterministic DAPC as masked reconstruction with additional regularization with predictive information). What is interesting here is that the latent space has a learnable structure that enforces high predictive information in the time axis, which is not considered in the standard VAE framework (and can not be modeled by standard Gaussian priors).
> * We added Table 6 and Figure 6 to show that we did optimize over the parameter k. We selected k=4 after tuning by R2 scores and showed the plots in the main paper. We also extended the description of the model architectures we use in Appendix B (the paragraph starting with “More specifically,...”). We will release the code for reproducing our experiments.
> * You are right that CPC maximize mutual information between current representation and future inputs (or sometimes shallow features of inputs) and thus by the data processing inequality, *implicitly* maximizing an *upper bound* of information between high level representations. However, we propose to explicitly maximizes the information between high level representations (the information estimate is in fact *exact* for Gaussian distributions), while having another regularization (masked reconstruction) that maximizes information between current input and current representations. Our results indicate that the explicit trade-off between the two is advantageous. We also added a paragraph in related work section to discuss this (the paragraph starting with “Note that by the data …”)

---

> > ### Comment · AnonReviewer4 · 2020-11-23
> > **Reply**
> >
> > Thank you for your reply. I have updated my rating to reflect the improvements made to the paper. Nonetheless, I still have two concerns:
> >
> > * Seeing the improvement that regularization through PI can bring over full reconstruction and masked reconstruction (Table 5, Figure 5, Table 7) is very interesting, and I think could be a bigger selling point for the paper. I believe that describing DAPC as a contrastive method that “can be computed without negative sampling”, while it still relies on a reconstruction loss to avoid degenerate solutions seems unnatural. In my opinion, the purpose of this perspective should be to motivate the usage of MI between different representations and to provide a broader view of existing representation learning methods. Ultimately, interpreting PI as a regularization on the latent space of an autoencoder could make the paper and its contributions easier to follow and highlight.
> >
> > * It is interesting to see the performance of CPC for different choices of k. However, it still remains unclear how much hyperparameter tuning has been done for the different methods. If they have been tuned to different extents, their comparison might not be fair.

---

> > > ### Author Response · Authors · 2020-11-24
> > > **DAPC is not a contrastive method and Model tuning is exhaustive**
> > >
> > > Thanks for the response. Further clarifications:
> > > 1. **(a)** Our method is NOT a “contrastive method without negative sampling”, and our method does not involve a contrastive loss. Rather, our method uses an estimator of predictive information (MI between past and future window) that does not require negative sampling. Using PI alone as the learning objective often achieves reasonable performance (see Figure 2, Table 1, Figure 5, Table 5), yet the latent space can lose information of the inputs and can be improved with a reconstruction loss; this is what we meant by “degeneracy”, and we do not refer “degeneracy” to difficulty in numerical optimization.  We were just trying to compare DAPC with CPC-like methods from a high-level to give the readers a big picture and stress on the key differences. DAPC is only similar to contrastive learning in that both model mutual information. Beyond this point, DAPC is quite different from contrastive methods. **(b)** Our loss function has 2 parts---PI-based loss and reconstruction-based loss---and either of them can be seen as the regularization to the other. AFAIK, we are the first to introduce PI for deep representation learning, and we focus more on it since the readers might be less familiar with it.  **(c)** We agree that PI can potentially be useful in a variety of methods, e.g., as we show for deep generative models in Sec 2.3.
> > >
> > > 2. For Sec 4.1 and 4.2, we used the best tuned DCA parameters from the original paper (Clark et al. 2019), and we further tuned the parameters like window size T based on a grid search for the applications we had. For these experiments, CPC has been extensively tuned as shown in the revision (Table 6 and Figure 6). For Sec 4.3, we mostly quote numbers from prior work that experimented with the same setup, including the wav2vec series which mainly implements the CPC loss. Wav2vec series however, use larger model architecture and much heavier tuning, see our general response. In contrast, we only carefully tuned PI only and MR only methods given that they are components of DAPC, and the trade-off parameter for them in the final loss. The comparisons are quite fair.

---

> > > > ### Comment · AnonReviewer4 · 2020-11-25
> > > > **Reply**
> > > >
> > > > Thank you for the clarification.
> > > >
> > > > 1) I would suggest including a discussion of this in the paper - for me, it was not clear that this is not a contrastive method. Coming from a contrastive learning background, phrases such as "can be computed without negative sampling" immediately create a mental connection to methods such as BYOL for me, making it hard to see this method as a non-contrastive method if not called out as such explicitly. Additionally, I believe that this would highlight the contribution of the paper more clearly.
> > > >
> > > > 2) Likewise, I would suggest including this discussion in the paper as well, as it will increase the expressiveness of the experiments.

---

### Official Review · AnonReviewer3 · 2020-10-28
**Decent analysis; weak experiments**

**Rating:** 6
**Confidence:** 4

**Review:**

This is an interesting paper with several proposed theories on how to improve self supervised learning objectives through a "predictive information" type objective, combined with a masked reconstruction loss. The hypothesis is interesting, and has some benefits, specifically that it does not require contrasting with negatives unlike many recent SSL methods.

Most experiments performed on the speech recognition task. While the theory and discussion looks plausible, the experiments are somewhat on the weak side. Specifically:

1) the differences in word error rates in the ablations are pretty small. The authors claim to run 3 seeds for each result and report the mean - can we see the standard deviation?
2) why not compare to wsj results from cpc-style training such as wav2vec and vq-wav2vec?
3) eval on Librispeech is only on the clean set. Why not show results on noisy, where differences between techniques should be somewhat more apparent?
4) there are newer models to compare against, e.g. wav2vec 2.0
5) the authors claim that they perform worse than e.g. BERT + vq-wav2vec because their model is smaller. But they use standard transformers as their backbone - it should be straight forward to scale this up and have apples to apples comparison.
6) this is positioned as a general SSL technique - what about experiments in other modalities like nlp or vision?
7) is "DAPC" just the PI loss without masked reconstruction? (i am not sure if "shifted recon" just adds the shifting, or it adds the entire reconstruction dimension). I think that is the case, but i am not 100% sure - maybe its worth to make this more clear.

Overall I liked the premise of the paper but unfortunately the experiments left me unconvinced in the value of this approach and its various components

Update:
thanks for your reply. i remain not fully convinced of the improvements with the proposed method and i look forward to additional experiments in NLP. i do think this approach is valuable for additional study however and updated my rating to reflect this

---

> ### Author Response · Authors · 2020-11-20
> **std added and some clarifications on your concerns**
>
> 1. While the differences between the baseline approaches (PI only, MR only) is not very large, the performance of the proposed method is significantly better (in relative improvement since our baselines are relatively strong). STD has been added to table 1 and 2.
>
> 2. We mainly use WSJ dataset for ablation study and model tuning, and there we tried to use the same implementation to rule out other factors. We added Table 9 reporting other methods’ performance for the same setup. Other models like wav2vec and vq-wav2vec didn’t report the numbers for the WSJ scenarios we used (81h pretrain + 81h finetune) since they pretrained on Librispeech. Figure 2 in wav2vec’s paper shows wav2vec’s performance in the plot but no specific numbers are reported. Nevertheless it is not hard to infer that wav2vec’s numbers are worse than DAPC’s.
>
> 3. For Librispeech test_other, we spot checked decoding results for a few models and observed the performance on test_clean and test_other are quite consistent, in the sense that relative merits between methods are maintained.
>
> 4. & 5. See general comment. We can not do apples to apples comparison due to limits on computational resources.
>
> 6. Our generality is also shown in the other domains: Lorenz Attractor, temperature dataset, hippocampus study and motor cortex dataset. We plan to apply our methods to NLP in the future.
>
> 7. DAPC is the combination of predictive information (PI) and masked reconstruction (MR). Table 1 and 2 are showing further variations of PI and MR, multi-scale PI and shifted MR. But DAPC itself also comprises PI and MR. We have clarified this in the text in the revised version.

---

### Official Review · AnonReviewer2 · 2020-10-29
**A neural extension of Dynamical Components Analysis**

**Rating:** 5
**Confidence:** 4

**Review:**

This paper builds on the prior work of Dynamical Components Analysis (DCA) which maximizes the mutual information between past and future temporal windows around the current time step, referred to as the the Predictive Information (PI) loss.
In this paper, the PI loss is used to train a neural encoder that learns continuous latent representations of input sequences. The PI loss is regularized to have orthogonal latent space. It is further improved by summing the PI loss applied at multiple scales, and by adding a masked reconstruction (MR) loss. The paper presents results on three domains, with speech recognition as the main one.
The paper presents a nice extension of DCA, as well as a probabilistic interpretation motivated by the variational autoencoder (VAE) framework. It is clearly written with good citations of previous work.

On the other hand, The experimental section requires more work. Here are some directions for improvements:
1) Adding the MR loss, combines another effective pre-training mechanize from previous work. I'm assuming that in tables 1 and 2, DAPC (written alone) refers to models optimizing the PI loss without MR (please correct me if this is not true). If my understanding is correct, the PI loss doesn't require MR to avoid degenerate solutions. Further analysis for the ASR models would help the readers understand different cases when the MR is required.
2) Following on the previous point, table 1 is missing the MR only results in the upper section, and missing the DAPC only in the section section. This is important to understand the relative contribution of each pre-training loss.
3) In table 2, the reported results are using a 30M parameter model compared to larger previously published models (~150M parameters). Given that the results of the proposed model is worse than those larger models, it is not clear why the authors didn't pre-train larger models of similar capacity for fairer comparison to their proposed approach. It is known that pre-training larger models yield better final representations for downstream tasks. Would it be the case for DAPC?
4) The pre-training on librispeech only is done for a 1 epoch. This seems pretty short compared to prior pre-training work on the same dataset (correct me if this is not accurate). Do the learned representations using DPAC stop improving after 1 epoch of pre-training? If possible, please share results of pre-training for larger number of epochs for the DPAC only, MR only, and combined loss.

---

> ### Author Response · Authors · 2020-11-20
> **Missing numbers added and some clarifications on your concerns**
>
> Thanks for recognizing some virtues of our paper. Some of our clarifications to your concerns:
>
> 1. DAPC is the combination of predictive information (PI) and masked reconstruction (MR). Table 1 and 2 are showing further variations of PI and MR, multi-scale PI and shifted MR. But DAPC itself also comprises PI and MR. We have added MR only and PI only results in Table 1. Please check it out and the results show that PI alone is not good enough. For the other 2 applications, we also added experiments in Table 5, Figure 5, Table 7 to show PI can greatly boost the performance of different reconstruction schemes while on the other hand, PI alone fails to provide meaningful results. We would appreciate it if you can take a look at the tables and figures.
>
> 2. In Table 1, we add PI-only and MR-only results in the upper section and DAPC-only results in the lower section.
>
> 3. Please refer to the general comment. We can not do apples to apples comparison due to limits on computational resources.
>
> 4. This is a subtle point. Any pretraining method uses some objective to learn useful features, which eventually is not aligned with downstream ASR and thus a stopping criterion/model selection strategy is needed. Another complication for us comes from the ESPNet asr pipeline we use, which combines attention and CTC with interplay between the two. On WSJ, we found that attention and ctc prefer different pretraining epochs; attention benefits from longer pretraining (e.g., 50 epochs) than CTC (e.g, 5 epochs). Thus for the hybrid attention + ctc model we pick the epoch that leads to overall best dev WER performance.  For the combined DAPC loss, in the pretrain on 81 hours and finetune on 15 hours setup (and a single random seed), pretraining epoch 1 leads to dev WER 12.2%, epoch 3 gives 12.0%, epoch 5 gives 12.4%, and epoch 20 gives 12.4%.
> On Librispeech the best pretraining epoch is 1, but this is partly because we used a quite small batch size (up to 32 utterances), and we actually have done 30K updates for pretraining.

---

### Official Review · AnonReviewer1 · 2020-10-30
**Accept**

**Rating:** 7
**Confidence:** 3

**Review:**

The paper contributes to the growing body of work on self-supervised representation learning approaches. It appears to have a strong theoretical foundation based on the concept of predictive information. The results seem competitive and there are visible efforts to place the method in the context of existing work.

Pros:
- Strong theoretical foundation, well explored.
- Competitive results in multiple contexts.
- Mostly clear with adequate detail for the method and experiments.
- Originality: combining DCA estimation of PI with additional regularization terms + shifted masked reconstruction.

Cons:
- Although there is plenty of discussion of related work and how the current method fits in with existing approaches, it would be helpful to tie these other approaches back more explicitly to the compared methods in Fig 3 / Tab 1 / Tab 2. Perhaps it would be possible even to add a table comparing the methods discussed across various dimensions (e.g. discriminative/generative, contrastive, mutual information entities [representation vs input, past vs future], model size, and other important characteristics in which they differ).
- It is not clear whether the results for other published methods are taken from their respective publications or reproduced. If taken from publications, are the models used comparable?
- I would like more discussion on the pros and cons of using masked reconstruction versus full reconstruction (as in a canonical auto-encoder) as a way to avoid learning degenerate representations.

Overall, I think this paper would be of interest to the community.

---

> ### Author Response · Authors · 2020-11-20
> **Your suggestions have been incorporated in the revision**
>
> Thanks for your constructive suggestions. Here are some of our clarifications:
>
> 1. We have added a table 10 to show the key properties of the major methods we compare against in the ASR pretraining task. Some of these methods may have variations with different properties and we list the original versions. Besides variants of our methods and CPC, we have used PCA, SFA and DCA for the other two tasks, which are deterministic and non-contrastive.
>
> 2. Most results are quoted from the same scenarios in corresponding papers directly, as in Table 2 for LibriSpeech and Table 9 for WSJ. Perhaps except for the wav2vec series which is clearly larger, models of compared methods have sizes of the same scale and similar architecture (which can also be inferred from the computational resources used to train them).
>
> 3. We have added table 5 and 7 for more observations on full reconstruction. Full reconstruction is useful in most scenarios but it is outperformed by masked reconstruction in general.
> It has also been observed for the ASR pretraining task that the full reconstruction task is too easy to learn useful features (see Wang et al 2020, section 4.2, second paragraph), since the powerful encoder gets to see all the input context. On the other hand, masked reconstruction forces the model to learn contextual information.

---

### Author Response · Authors · 2020-11-20
**Paper revision**

We thank the reviewers for their constructive comments. We have updated the pdf to include more related work, and have included more ablation studies for the Lorenz attractor, forecasting tasks, and the Wall Street Journal datasets. While Reviewers 2, 3 had concerns on the model size for ASR experiments on Librispeech and perhaps comparison with the wav2vec series, we would like to point out that most institutes do not have as many computational resources for extensive experimentation with gigantic models in the speech domain. As you can see, the wav2vec series have regularly used 64 or more GPUs for training a single model, and their language models can be very large for best performance (e.g., wav2vec 2.0 used a transformer LM with 20 layers and 16 attention heads); such model sizes make tuning extremely time consuming. Our models have reasonable sizes that allow most thorough analysis with the resources we have, and our baselines already outperform most existing methods with similar model sizes.

We would like to reiterate that the purpose of our paper is to propose a general representation learning method for sequence data, beyond just the speech domain, and we hope the reader will also appreciate the other experiments, i.e., Lorenz attractor and the forecasting tasks, which were used in previous work.

We have also replied to each reviewer’s comment separately.

---

### Decision · Program_Chairs · 2021-01-07
**Final Decision**

**Decision:**

Accept (Poster)

**Comment:**

The paper combines a few different ideas for representation learning on sequential data and is able to achieve competitive WER on the Librispeech ASR dataset. I appreciate the fact that the authors engaged with reviewers and tried to improve the paper. While I get a sense that the final system has many moving parts, I believe the paper meets the bar for acceptance at ICLR.